# Lung Cancer Subtyping: A Short Review

**DOI:** 10.3390/cancers16152643

**Published:** 2024-07-25

**Authors:** Farzana Siddique, Mohamed Shehata, Mohammed Ghazal, Sohail Contractor, Ayman El-Baz

**Affiliations:** 1Department of Bioengineering, University of Louisville, Louisville, KY 40292, USA; farzanasiddique459@gmail.com (F.S.); mohamed.shehata@louisville.edu (M.S.); 2Electrical, Computer, and Biomedical Engineering Department, Abu Dhabi University, Abu Dhabi 59911, United Arab Emirates; mohammed.ghazal@adu.ac.ae; 3Department of Radiology, University of Louisville, Louisville, KY 40202, USA; sohail.contractor@louisville.edu

**Keywords:** lung cancer subtypes, transcriptomics, proteomics, metabolomics, immunohistochemistry

## Abstract

**Simple Summary:**

Lung cancer is the most commonly diagnosed and most lethal cancer. Since healthcare outcomes have improved with the advent of targeted therapy, it is incumbent upon treating physicians to precisely identify the particular histological subtype. This paper summarizes studies exploring immunohistochemical and “omics” techniques to highlight diagnostic biomarkers for lung cancer subtyping. A comprehensive discussion along with an elucidation of challenges and future direction for further progress are also touched upon.

**Abstract:**

As of 2022, lung cancer is the most commonly diagnosed cancer worldwide, with the highest mortality rate. There are three main histological types of lung cancer, and it is more important than ever to accurately identify the subtypes since the development of personalized, type-specific targeted therapies that have improved mortality rates. Traditionally, the gold standard for the confirmation of histological subtyping is tissue biopsy and histopathology. This, however, comes with its own challenges, which call for newer sampling techniques and adjunctive tools to assist in and improve upon the existing diagnostic workflow. This review aims to list and describe studies from the last decade (n = 47) that investigate three such potential omics techniques—namely (1) transcriptomics, (2) proteomics, and (3) metabolomics, as well as immunohistochemistry, a tool that has already been adopted as a diagnostic adjunct. The novelty of this review compared to similar comprehensive studies lies with its detailed description of each adjunctive technique exclusively in the context of lung cancer subtyping. Similarities between studies evaluating individual techniques and markers are drawn, and any discrepancies are addressed. The findings of this study indicate that there is promising evidence that supports the successful use of omics methods as adjuncts to the subtyping of lung cancer, thereby directing clinician practice in an economical and less invasive manner.

## 1. Introduction

Lung cancer accounts for 11.4% (2.21 million) of the total cancer incidence globally, as reported by (GLOBOCAN 2020). Out of the 10 million annual cancer-related deaths worldwide, 18% (1.79 million deaths) are attributable to lung cancer [1]. Lung and bronchus cancer is the third most common cancer and leading cause of cancer mortality [2]. Although lung cancer has over 50 histomorphological subtypes, in broader terms, there are two main types of lung cancer: non-small cell lung cancer (NSCLC) and small cell lung cancer (SCLC), with NSCLC being the more common type, constituting 85% of total lung cancer cases [3,4]. The remaining newly diagnosed lung cancer cases are SCLC, which generally has a poorer prognosis, with high relapse rates and low overall survival (OS), even after therapy. There are three major histological subtypes of NSCLC: adenocarcinoma (ADC) (38.5% of all lung cancers), squamous cell carcinoma (SCC) (20%), and large cell carcinoma (LCC) (3%) [5]. According to the National Cancer Institute, the financial burden of cancer care for lung cancer in the U.S. in 2020 was $23.8 billion USD [6]. 

With the advent of matched targeted therapy for NSCLC, lung cancer treatment has now progressed beyond the conventional chemotherapy- and radiotherapy-based options [7]. Various drugs targeting EGFR, ALK, ROS1 mutations, and VEGF are being tested and used in the clinical setting after carefully tailoring the therapy to the patient’s specific tumor type [8]. In some biomarker-defined subsets, these targeted therapies are more effective than cytotoxic chemotherapy [9]. Between 2013 and 2016, there was a greater decline in mortality rates than in incidence rates for NSCLC, which coincides with the timing since targeted therapy was approved [10]. Advances in immunotherapy, e.g., PDL1 inhibitors, are also underway [10]. To reap the maximum benefit from these advances in molecular targeted therapy and immunotherapy, it is imperative to properly subtype NSCLC [10,11]. In the case of SCLC, although there has not been a demonstrable increase in survival rates over the years, we can be hopeful that changes made to the morphological classification will direct new trials investigating targeted therapies for different subtypes of SCLC [11].

Conventionally, tissue biopsy has been deemed the gold standard of lung cancer confirmation [12]. But the procedures required to obtain tissue for biopsy, e.g., bronchoscopy, mediastinoscopy, thoracoscopy, and surgical excision, are expensive and invasive [13]. Furthermore, for histological classification, samples need to have a sufficient amount of tissue, which creates an avenue for possible repeat testing [13]. These limitations warrant the utilization of biological samples that can be attained in a less invasive and more economical manner, thereby benefiting those patients unable to undergo procedures for solid biopsy. Over recent years, liquid biopsy samples (plasma, serum, saliva, sputum, cerebrospinal fluid, pleural fluid, etc.) are gaining increasing attention in the diagnostic workflow of cancer [14,15]. This technique of sample procurement along with the detection of qualitative and/or quantitative indicators of the state of the normal or diseased body (biomarkers) could potentially improve existing diagnostic pathways [16]. Currently, in the context of lung cancer diagnostic biomarkers, the efficacy of circulating tumor cells (CTCs), circulating tumor DNA (ctDNA), exosomes, microRNA (miRNA), proteins, and metabolites is being explored [15].

The 2015 World Health Organization (WHO) Classification of Tumors of the Lung, Pleura, Thymus, and Heart integrated immunohistochemistry (IHC) into the classification flow for lung cancer and emphasized the importance of molecular characterization in 2021 [13,17]. IHC is a highly versatile tool to localize proteins in biological samples that allows for greater diagnostic accuracy in terms of subclassification and is also now routinely used as a part of the clinical workup for lung cancer, specifically in the case of small biopsies, cytological specimens, and poorly differentiated tumors [18]. As an added benefit, a diagnosis of NSCLC not otherwise specified (NSCC NOS) can be avoided in 90% of cases by using IHC [17]. But this method also comes with its pitfalls. IHC markers are incapable of distinguishing certain primary lung carcinomas from secondary metastases from other sites [19]. Furthermore, IHC is an arduous process, and IHC markers cross-react with normal lung tissue [20]. Ideally, an effective diagnostic biomarker should be present only in diseased patients and absent or found in small quantities in normal controls, have an association with tumor burden, be easily measurable with reliable techniques, and be economical while also showing high sensitivity and specificity [21]. This is why the addition of “omics” markers could potentially fulfill these goals. 

Transcriptomics is the study of a specific nucleic acid called RNA, and since it revolves around post-transcriptional events, it can screen out changes not discovered by genomic techniques [22]. It focuses on the effects that environmental factors could have on the transcriptome [23]. Furthermore, transcriptomic methodologies can quantify and evaluate the expression of microRNAs, which are small, non-coding stretches of nucleic acid that have regulatory functions [24]. Common techniques used are reverse transcription polymerase chain reaction (RT-PCR), quantitative (real-time) reverse transcriptase PCR (qRT-PCR), and microarrays [24]. The utilization of transcriptomics could add value in the field of precision medicine [23]. 

Proteins are the final determinant of a cell’s phenotypic characteristics [25]. They are extremely dynamic because of the multitude of changes that they go through due to factors such as post-translational modification and drug treatment [25]. A comprehensive analysis of the protein profiles of an organism is known as proteomics [26]. Proteomics can be subdivided into the following types: sequence, structural, functional and interaction, and expression proteomics [27]. Various techniques have been developed to build proteomes. Gel-based (one-dimensional (1D) gel electrophoresis, two-dimensional (2D) polyacrylamide gel electrophoresis, and difference gel electrophoresis (DIGE)), gel-free (liquid chromatography coupled to mass spectrometry (LC/MS)), and antibody-based (enzyme-linked immunosorbent assays (ELISAs)) proteomics methods are frequently employed to identify proteins in biological samples [28]. Proteomics research on lung cancer would benefit greatly from the development of LC/MS-based technologies, which can handle more than 100 samples per day with high precision [11]. To summarize, proteomics holds promise in the field of diagnostics, therapeutics, and prognostics for lung cancer. 

Metabolomics is complementary to transcriptomics and proteomics in the sense that it examines the consequences of the interactions of DNA, RNA, and proteins [29]. A vast number of small molecules can be studied by quantitative and qualitative analysis, and it is a true representation of the present activity of a biological system [29,30]. Metabolomics is an important tool to investigate lung cancer, as malignant cells show changes in certain metabolic pathways, e.g., upregulation of glycolytic and phospholipid pathways, as well as the TCA cycle and fatty-acid synthesis. The metabolomes of different histotypes are also distinct [31]. Traditionally, techniques such as gas chromatography–tandem mass spectrometry (GC-MS), nuclear magnetic resonance (NMR), and high-performance liquid chromatography UV detection (LC-UV) were used for metabolic profiling, with limitations like the inability to cover a large number of substrates. These techniques have since evolved into tandem mass spectrometry (MS/MS) and high-resolution tandem time-of-flight mass spectrometry (TOF/TOF), with a resultant increase in the number of metabolites that can be studied [32]. 

In this review, we have aimed to compile and describe various studies that investigate the role of transcriptomics, proteomics, metabolomics, and immunohistochemical markers in solid and liquid biopsies to highlight diagnostic biomarkers for lung cancer subtyping. Integrated approaches have also been discussed. 

## 2. Search, Inclusion, and Exclusion Criteria

Studies were chosen after carrying out searches in the reputed search engines PubMed, Google Scholar, and ResearchGate. The following keywords were used, either alone or in combination: “Lung cancer”, “NSCLC”, “SCLC”, “ADC”, “SCC”, “omics”, “Transcriptomics”, “Proteomics”, “Metabolomics”, “Immunohistochemical Markers”, “Diagnosis”, “Histology”, and “Subtyping”. Studies included were limited to the last decade. The following inclusion criteria were applied: English-language articles, original research, human studies, in vivo studies, studies involving adult participants, and studies on primary lung cancer and diagnosis. Articles such as systematic reviews, reviews, book chapters, in vitro studies, animal studies, and research involving aspects other than diagnosis, i.e., treatment, were not selected. Please see Figure 1 for more details.

## 3. Immunohistochemical Markers

In a study conducted by Argon et al. [33], a total of 120 specimens previously diagnosed histologically were immunohistochemically stained with TTF-1, CK5/6, and p63. Out of 72 SCC cases, 70 were CK5/6 positive, while all ADC cases turned out to be negative for the same marker. P63 positivity was seen in all squamous cell carcinomas and in four adenocarcinomas. All 72 cases of SCC were negative for TTF1, while 19 cases of ADC were positive. None of the 17 NSCLC cases later confirmed to be SCC was positive for either marker, whereas all of the NSCLC cases later confirmed to be ADC were positive for TTF1 and negative for CK5/6. They found that TTF1 could be considered the most reliable marker to differentiate between SCC and ADC, with a sensitivity and specificity of 100%. CK5/6 and p63 could be used as adjuncts to TTF1, with a sensitivity and specificity of 97% and 100% and of 100% and 87%, respectively.

Kim et al. [34] conducted a study with the aim of validating a variety of known immunohistochemical markers and study additional markers (i.e., Napsin A, CK5/6, Epithelial-related antigen (MOC-31), CD141, p27, CEA, TTF-1, p63, Cyclooxygenase 2 [COX-2], high-molecular-weight CK [HMWCK], and Rb protein) to accurately differentiate between ADC and SCC with obvious squamous or glandular differentiation in surgical resectates. In this study, Napsin A was found to have a greater sensitivity and specificity than TTF1 (81% and 100% vs. 70% and 98%) for determining ADC. Since not all cases of ADC were positive for either marker, they concluded that a combined panel of Napsin A and TTF1 could serve as a better alternative to using either alone for the accurate diagnosis of ADC. CK5/6 stained 90% of SCC and only 4% of ADC cases (4%), with a sensitivity of 90% and specificity of 96%. It was inferred that CK5/6 was a more efficient marker than P63 for ADC. CD141 was also deemed a prospective novel biomarker for SCC with a high specificity, although it had a low sensitivity. They found out that P27, Rb, HMWCK, MOC-31, COX-2, and CEA might have limited value in the diagnosis of ADC. This study elected a combined panel of Napsin A, TTF1, CK5/6, and p63 as the best option to discriminate between ADC and SCC. 

Yassin et al. [35] explored the role of the immunohistochemical expression of ALDH1A1 in spectral lung lesions and also ascertained any correlation of such expression with certain pathological parameters. The study included 105 specimens from healthy, non-cancerous, and cancerous lung tissue. Sections were then further classified into NSCLC (ADC and SCC), SCLC, and inflammatory pseudotumors. ALDH1A1 positivity was found in 83% of ADC cases; furthermore, they determined that ALDH1A1 expression negatively correlated with the histological grade. As for SCC, 100% of the cases displayed positive expression of ALDH1A1, and similar to the case of ADC, ALDH1A1 expression had a negative correlation with histological grade. ALDH1A1 positivity was demonstrated in only 50% of SCLC cases, although they were found to have a more homogenous distribution in comparison to NSCLC sections. 

Nishino et al. [36] endeavored to establish a two-antibody cocktail (from a panel of TTF1, NapA, CK5/6, and P40) for NSCLC subtyping with the aim of maximizing the capacity of cytology/small biopsy specimens in molecular studies. The study was conducted in multiple phases. TMAs were formed from FFPE sections of wedge and lobectomy specimens of lung ADC and SCC. In the first phase of the study, each immunohistochemical marker was examined individually using TMAs, and the intensity and distribution of expression were also noted. This process was repeated with a dual-marker panel in the second phase. The final phase involved the staining of preoperative cytology and small biopsy specimens that could not be accurately subtyped due to poor differentiation, lack of preservation of tissue, or poor tumor cellularity. Matched surgical resection specimens were used for reference diagnosis. In the first phase, P40 displayed greater specificity than CK5/6 for SCC (94% vs. 59%), with similar sensitivity (100%). TTF1 had similar specificity (99% vs. 91%) and sensitivity (94%) for ACA. ROC analysis determined a significantly higher cutoff value for CK5/6 than p40, although cutoff values were similar for TTF1 and Nap A. After completion of the first phase, a dual panel of p40/Nap A was selected as the most fitting panel to graduate on to the second phase. As a whole, the Nap A+/p40− combination showed a specificity of 100% and sensitivity of 94% for ADC, whereas the Napsin A−/p40+ immunophenotype was 100% specific and 100% sensitive for SCC. In the final phase of the study, the Nap A/p40 cocktail successfully identified 88.8% of the cytology and small biopsy specimens on a single slide as ADC, SCC, or NSCLC-NOS. The study concluded that a cocktail of Nap A/p40 can be used to accurately subtype preoperative cytology and small biopsy specimens as ADC/SCC. For specimens in which tissue is scant, a stepwise algorithm beginning with this cocktail can be employed in order to save tissue for further molecular tests. It can also be used to confirm the diagnosis of well- to moderately differentiated tumors in which diagnosis is obstructed by crush artifact or paucicellularity.

Wei et al. [37] studied a total of 395 (including 365 lung cancer and 30 normal lung) samples in order to determine the clinicopathological significance of CDK5 expression in lung cancer. TMAs were constructed, which were later stained with CDK5. CDK5 positivity in lung cancer samples highly exceeded that in normal lung tissue (51.5% vs. 20%). The positive rates were found to be higher in advanced cases, cases of lymphatic metastases, and cases with a poorer histological grade. ROC curve analysis revealed an AUC of 0.685 for CDK5.

Guo et al. [38] conducted a study using dual-marker stains (p40/Nap A and CK5/6/TTF1) with the objective of differentiating ADC and SCC in single sections of lung cancer tissue. A total of 58 lung cancer specimens were prepared and stained with the dual antibody panel, with single-marker-stained sections kept as controls. Upon single-marker staining, 100% of SCC cases showed p40 (nuclear) and CK5/6 (cytoplasmic) positivity, 93.8% of the ADC cases showed TTF1 (nuclear) positivity, and 87.5% cases showed Nap A (cytoplasmic) positivity. The p40 and Napsin A dual-marker set showed 100% nuclear positivity and cytoplasmic negativity with SCC cases, and the nuclei stained negative while the cytoplasm stained positive in 87.5% ADC cases, whereas CK5/6 and TTF1 dual staining exhibited 100% cytoplasmic positivity and nuclear negativity in SCC cases, and the nuclei stained positive while the cytoplasm stained negative in 93.8% ADC cases. The single- and dual-marker panels revealed identical results, which led the investigators to conclude that dual-marker staining could be an economical and simple novel approach to differentiating subtypes of NSCLC and could be an alternative to the more conventional single-marker IHC panels.

Zyl et al. [39] carried out a study with the objective of using immunostaining to distinguish between SCC and ADC. They selected a total of 271 fine-needle-aspiration cell blocks (FNABs) and formalin-fixed paraffin-embedded (FFPE) tissue block specimens that were stained with TTF-1, Napsin A, CK5, and P63. TTF-1 and Napsin A showed high sensitivity (99.0% vs. 91.9%) for ADC of the lung. Although these markers had similar PPVs and NPVs, Napsin A displayed a higher specificity than TTF-1 (90.8% vs. 62.8%). For SCC cases, CK5 and P63 had similar sensitivity rates (95.4% vs. 97.9%) and NPVs, but CK5 exhibited a greater specificity, PPV, and accuracy than P63. 

Ao et al. [40] attempted to examine the usefulness of a newly designed triple-marker panel for the accurate diagnosis of different NSCLC subtypes. Tissue microarrays were built from surgically resected specimens (n = 200). The study aimed to compare the diagnostic abilities of the triple-marker panel (consisting of TTF-1, Napsin A, and p40) to those of conventional single markers alone. The study revealed the following findings: In regard to ADC, the triple marker showed marginally higher sensitivity than TTF-1 and Napsin A alone (93.5% vs. 85.7% and 89.6%). Specificity was found to be lower for the triple marker than for Napsin A alone (77.5% vs. 90.0%). There was no statistically significant difference between the triple and single panels. For SCC, the triple-marker panel achieved a slightly better sensitivity than P40 and Ck5/6, but it a bit lower than that of P63. In terms of specificity, the triple-marker panel was the highest when compared to the others alone. There was no statistically significant difference between the panels. Overall, the study found that the triple-marker panel had similar sensitivities and specificities when compared to individual panels. This would be very helpful for the accurate subclassification of NSCLCs and would also aid in tissue preservation for further necessary molecular testing.

Kawai et al. [41] investigated the role of five immunostains (TTF1, Napsin A, p63, CK5/6, and desmocollin-3) on touch imprints from 215 surgically resected lung cancer specimens for the diagnosis of SCC and ADC. The goal of this study was to compare between conventional cytological staining and immunohistochemistry. Although only 73% of the IHC samples matched with their cytological counterparts, Napsin A was 80% sensitive and 97% specific when it came to differentiating between ADC and SCC, while for P63 the values were 84 and 98%, respectively. This study concluded that Napsin A and P63 could be very helpful in the histological categorization of NSCLC.

In their study, Tran et al. [42] evaluated five different immunohistochemical markers and also aimed to compare the different commercially available antibody clones for lung cancer diagnosis. Tissue microarrays (TMAs) were constructed and consequently stained immunohistochemically and analyzed from a large sample of 557 resected specimens from two different cohorts. They were able to distinguish CK5 and p40 as good IHC markers for SCC identification and the exclusion of NSCLC of a non-squamous type. The isotype p40 showed superior diagnostic performance to P63. NAP A and TTF1 were found to be good markers to eliminate SCC but had a subpar capacity for the identification of NSCLC of a non-squamous subtype, although this limitation was alleviated to a degree by combining the two markers. A combined panel of ck5 and p40, however, did not show better diagnostic abilities than singular panels. In regard to clones, the TTF-1 clone SPT24 had a slightly higher sensitivity and lower specificity than clone 8G7G3/1, but NAP A and P40 clones displayed similar diagnostic performance. 

With the aim of exploring the impact of ALDH1 and CD133 on the histotypes of lung cancer, Roudi et al. [43] applied immunohistochemistry to TMAs constructed from 133 lung tumor samples. Their findings indicated that ALDH1 levels were higher in NSCLC cases than SCLC cases. Furthermore, of the NSCLC cases, SCC had the highest ALDH1 expression. Similarly, greater levels of CD133 were found in NSCLC cases compared to SCLC cases, although significantly different levels were not determined between the different NSCLC subtypes. Further analysis revealed that only ADC and SCC cases exhibited high levels of both markers, while all other tumor types displayed high levels of one marker and lower levels of the other marker. Overall, this study found that ALDH1 and CD133 could be used as novel biomarkers for lung cancer subtyping. 

In another study, Roudi et al. [44] investigated the expression and clinical role that CD44 might play in lung cancer subtyping. TMAs built from 195 lung cancer samples were evaluated with IHC staining. NSCLC cases exhibited a greater expression of CD44 than SCLC, while among the NSCLC subtypes, SCC showed higher levels of CD44 compared to ADC. Higher levels of the same marker corresponded with higher-grade SCC tumors with poorer prognosis. Conversely, lower levels indicated well-differentiated ADC tumors. The researchers concluded that CD44 could potentially serve as a novel diagnostic biomarker and might also be used to formulate targeted therapies. 

Table 1 synopsizes studies exploring the role of immunohistochemical markers in lung cancer subtyping. The data are sorted according to the following characteristics: study, sample size/type, markers, methods, results, and conclusion. A highlights portion has been included in the footnote consisting of the key findings.

## 4. Omics Markers

### 4.1. Transcriptomics

Geng et al. [45] explored the performance of miRNAs as novel biomarkers for the diagnosis of NSCLC. After analyzing the blood samples of 50 individuals with quantitative real-time PCR (qRT-PCR) in a training set, a panel of five miRNAs (miR-20a, miR-223, miR-21, miR-221, and miR-145) graduated to the validation set for further testing on a larger sample. ROC analysis of the obtained data sets was then used to determine the predictive capacity of the MiRNAs. miR-20a, -223, and miR-155 were found to perform best among the candidates in the validation set, with AUCs of 0.89, 0.94, and 0.92. The authors stipulated that these five miRNAs could individually work as noninvasive biomarkers for early stage NSCLC, although their practical use warrants further confirmation.

Lu et al. [46] identified six miRNAs based on a study conducted on 1132 individuals. This set was chosen using qRT-PCR from a total of 723 screened by microarray. Two logistic regression models were set up and validated using two independent cohorts. Panel A displayed good performance in assessing the risk of lung cancer, while panel B (miR-17, miR-190b, and miR-375) had good discriminating power between SCLC and NSCLC (AUC 0.878 and 0.869 for the training and validation cohort, respectively). However, the further subdivision of NSCLC was not possible. The authors concluded that these panels could potentially serve as effective markers for the early diagnosis and subtyping of lung cancer.

Jin et al. [47] explored the role of miR-365 in the carcinogenesis of lung cancer. Microarrays were used to screen out miRNAs, and qRT-PCR was used to evaluate miRNA expression in the validation set on 78 macrodissected frozen lung tissue samples. miRNA-365 was found to be upregulated in SCLC and ADC and downregulated in SCC. Furthermore, this study found that miRNA-365 encourages cell division in SCLC, which is a novel finding. 

Yang et al. [48] investigated a set of eight circulating miRNAs to determine the best possible model for early NSCLC diagnosis and possible histological subtyping. They employed qRT-PCR to analyze the serum levels of these markers in training and testing sets. Logistic regression models identified a panel of four miRNAs (miR-146b, miR-205, miR-29c, miR-30b, and miR-337) as best suited for the early diagnosis of NSCLC. Furthermore, this panel was found to be more predictive (AUC 0.98 vs. 0.93) and more sensitive (99.1% vs. 90.3%) for ADC compared to SCC. Subsequent survival analysis also revealed a poorer prognosis for SCC (*p* = 0.0035) with higher levels of miR-146b, but not for ADC patients (*p* = 0.83).

Zhang et al. [49] worked with a group of miRNAs to determine whether differential expression could potentially help in the early diagnosis of NSCLC. Sixteen miRNAs were screened using microarrays, of which six were validated using qRT-PCR. NSCLC patients had significantly higher levels of miR-3149 and miR-4769.3p, and they performed well as diagnostic markers (AUC 0.830 and 0.735, respectively), although the values were found to be similar in both ADC and SCC.

In a study conducted by Powrozek et al. [50], they investigated miR-944 and miR-3662 expression as potential biomarkers for the type-specific diagnosis of lung cancer. They employed RT-PCR to examine the blood samples of individuals and used ROC analysis to determine the diagnostic capability of each. Both miRNAs were expressed at a higher degree in cancerous samples compared to the non-cancer cohort. Moreover, miR-944 was found to display greater accuracy for operable SCC (AUC = 0.982), and miR-3662 for ADC (AUC = 0.926). In short, this study indicates that miR-944 and miR-3662 could serve as potent biomarkers for lung cancer diagnosis in the future. 

Singh et al. [51] evaluated six miRNAs in blood samples from patients with ADC and SCC. They analyzed the expression of these non-coding RNAs using qRT-PCR and reported significantly higher expression of mir-2114 and mir-449c in AC and mir-2115 in SCC. The authors summarized that these markers could be effective in the differential diagnosis of NSCLC within the scope of noninvasive approaches. 

Kumar et al. [52] conducted a study to determine whether miRNAs could act as noninvasive diagnostic biomarkers for NSCLC. They quantified the expression of miRNAs in the sera of 115 individuals (75 NSCLC patients and 40 healthy controls) by RNA sequencing and validated a set of 10 differentially expressed miRNAs by RT-PCR. miR-15a-5p, miR-320a, miR-25-3p, miR-192-5p, let-7d-5p, let-7e-5p, miR-148a-3p, and miR-92a-3p were found to be lower in NSCLC patients. SCC patients exhibited lower levels of miR-375 and miR-10b-5p compared to controls, whereas there was no significant difference in these markers between ADC patients and controls. Overall, this study concluded that miRNA-based assays could assist in the diagnosis and prognosis of NSCLC.

In another study, Kumar et al. [53] analyzed the differential expression of three miRNAs in 161 tissue samples (29 tumor resections with paired healthy tissue and 103 tissue biopsies) employing TaqMan advanced miRNA assays. The assay revealed the significantly upregulated expression of miR-197-3p and miR-375-3p in tumor resectates compared to that in healthy controls and of miR-375-3p expression in tissue biopsy samples. miR-375-3p performed the best as a diagnostic marker with an AUC of 0.7491. Furthermore, miR-375-3p and miR-197-3p were found to be good differentiators of SCC and ADC.

Nadal et al. [54] aimed to assess how miRNA signatures could function as adjunctive diagnostic tools for NSCLC. They generated miRNA expression profiles from the sera from 72 NSCLC patients at surgery and 22 controls. A set of four miRNAs was then obtained after further validation in a separate cohort using qRT-PCR. This combined panel achieved an AUC of 0.993 after ROC analysis for the diagnosis of NSCLC. The authors stipulated that this miRNA signature could act as a clinically helpful tool in the determination of lung cancer in patients. 

Jin et al. [55] used tumor-derived exosomal miRNA profiles for the histological subtyping of NSCLC. They utilized RNA sequencing to create unique miRNA signatures in the sera of 46 stage I NSCLC patients and 42 healthy controls. ADC patients exhibited significantly upregulated levels of 11 and downregulated levels of 13 miRNAs, while SCC patients showed upregulated levels of 6 and reduced levels of 8 miRNAs. Further validation revealed that miR-181-5p, miR-30a-3p, miR-30e-3p, and miR-361-5p were specific for ADC and miR-10b-5p, miR-15b-5p, and miR-320b for SCC. A three-miRNA combined panel showed promise as a good diagnostic marker for NSCLC, ADC, and SCC with AUCs of 0.899, 0.936, and 0.911, respectively. 

Fan et al. [56] studied the expression of 12 miRNAs in two cohorts (94 NSCLC patients vs. 58 healthy individuals) using qRT-PCR. The miRNAs that were differentially expressed were then analyzed using fluorescence quantum dot liquid bead arrays for differentially expressed miRNAs in serum samples of 70 NSCLC patients and 54 healthy controls. A panel of six miRNAs (miR-16-5p, miR-17b-5p, miR-19-3p, miR-20a-5p, miR-92-3p, miR-15b-5p) was able to distinguish between the two groups. Overall, a panel consisting of miR-16-5p, miR-15b-5p, and miR-20a-5p performed as the best discriminant for NSCLC. The authors concluded that these six markers could be explored as an avenue for noninvasive NSCLC diagnosis. 

Saviana et al. [57] conducted a study to construct a miRNA classifier that effectively distinguishes SCLC from other histotypes of lung cancer. After a stepwise experimental flow employing techniques such as RNA sequencing (n = 33) on a training set and qRT-PCR (n = 111) on a validation set, they finalized miR-375-3p as the most competent discriminant for SCLC. Statistical analysis revealed AUC = 0.793 with 37.5% sensitivity but 100% specificity.

Table 2 provides a summary of studies describing transcriptomic markers with highlights.

### 4.2. Proteomics

Yu et al. [58] used gene expression programming to construct optimal biomarker joint models to classify lung tumors. A set of five biomarkers (CEA, NSE, CPR, LDH, and CYFRA 21-1) was measured in the sera of 180 subjects diagnosed with malignant lung disease using various techniques. Each individual biomarker was statistically analyzed using SPSS 19.0, and the data were compared using analysis of variance (ANOVA) statistical method. GEP was then employed to build three models, and the predictive power of each to distinguish histological subtypes of lung cancer was assessed. GEP 3 (CEA + NSE + CYFRA 21-1) displayed an accuracy of 94.8%, which was the highest among the three models. In conclusion, a panel consisting of CEA, NSE, and CYFRA 21-1 was found to be superior for classifying lung cancer.

Visser et al. [59] studied liquid biopsy-based decision algorithms to diagnose and distinguish lung cancers into NSCLC and SCLC. Eight protein tumor markers (CA125, CA15.3, CEA, CYFRA 21-1, HE4, NSE, ProGRP, and SCCA) were measured using electrochemiluminescent assays, and ctDNA mutations in EGFR, KRAS, and BRAF were determined using droplet digital PCR in the blood of 1096 patients suspected of having lung cancer. Multivariate logistic regression models were utilized to assess the performance of the tumor markers in combination or individually. The most informative markers to be incorporated in the multiparametric models were elected. CYFRA 21-1 had the best overall performance (AUC = 0.78) and highest sensitivity (25%) in identifying NSCLC, while ProGRP had an AUC of 0.86 and sensitivity of 40% in identifying SCLC, with a PPV of 100%. Moreover, a model combining CYFRA 21-1, CEA, ProGRP, and NSE was shown to perform better (AUC = 0.86) and with a higher sensitivity when compared to CYFRA 21-1 alone for NSCLC diagnosis, and a panel of CA125, CA15.3, CYFRA 21-1, NSE, ProGRP, and DNA was superior to individual TMs when diagnosing SCLC.

Korkmaz et al. [60] studied a panel of six tumor markers (progastrin-releasing peptide (ProGRP), squamous cell carcinoma antigen (SCCAg), cytokeratin 19-fragments (CYFRA 21-1), human epididymis protein 4 (HE4), chromogranin A (CgA), and neuron-specific enolase (NSE)) to determine the role they could play in the diagnosis and classification of lung cancer. They quantified the level of these markers in the venous blood of 99 lung cancer patients and 30 patients with benign lung disease using mass spectrometry. ROC analysis was used to evaluate the diagnostic performance of each. ProGRP levels were found to be greater (*p* = 0.009) in SCLC; CYFRA 21-1 and SCCAg levels were greater in NSCLC (*p* = 0.019 and *p* = 0.001, respectively). Furthermore, CYFRA 21-1 (*p* < 0.001, r = 0.394), HE4 (*p* = 0.014, r = 0.279), and CgA (*p* = 0.023, r = 0.259) levels were shown to exhibit a positive correlation with the stage of NSCLC. Regarding the distinction between the two histological subtypes, ProGRP achieved the best overall performance (AUC = 0.875).

A study by Wen et al. [61] aimed at constructing a set of combined tumor markers for the diagnosis of different lung cancer types by assessing a panel of 10 tumor markers. They measured the concentration of each in serum samples from 250 individuals with and without lung cancer. The diagnostic values were compared and contrasted using a chi-squared test. The CEA level (AUC: 0.812; 63.9% sensitive) was the highest related to ADC; CYFRA 21 (AUC: 0.847; 84.6% sensitive) and CEA (AUC: 0.804; sensitivity: 70.0%) were the best suited for SCC and NSE (AUC: 0.819; 69.0% sensitive), and CEA (AUC: 0.808; 60.7% sensitive) was the highest in regard to SCLC. 

Trulson et al. [62] employed a group of serum tumor markers (CYFRA 21-1, CA 125, CA 15-3, CA 19-9, CEA, NSE, ProGRP, SCC, and CA 72-4) to discriminate between the histological subtypes of lung cancer. A case control group of 490 patients with malignant and benign lung disease was studied. They used logistic regression to compare the differentiating capability of each tumor marker singly or in combination. Their findings indicate that ProGRP and NSE were superior for identifying SCLC and NSCLC (AUC 0.86 and 0.83, respectively). The performance improved with a two-marker panel (NSE and ProGRP) and a three-marker combination (NSE, ProGRP, and CEA), both yielding an AUC of 0.97. Low SCC and high CA15-3 values further subdivided NSCLC into ADC and SCC (both with an AUC of 0.75). The authors concluded that a combination of serum tumor markers could be useful in assessing different subtypes of lung cancer. 

Sua et al. [63] investigated a group of 93 individuals suffering from lung diseases to study the efficiency with which a set of five serum biomarkers could differentially diagnose the histological subtypes of lung cancer before tissue diagnosis. ROC analysis was utilized to compare the efficacy of the markers. The malignant lung disease cohort yielded significantly greater median levels of CYFRA 21-1, SCC-Ag, Pro-GRP, CEA, and NSE compared to the benign lung disease group. In terms of subtyping, the performance of individual markers was widely varied, and combined panels performed significantly better. For NSCLC, the panel of Pro-GRP, CEA, CYFRA 21-1, SCC-Ag, and NSE achieved the highest AUC of 81.1% (sensitivity = 74.5% and specificity = 78.8%), and CYFRA 21-1 + Pro-GRP + NSE performed best for SCLC (AUC = 97.3%, sensitivity = 88.8% and specificity = 98.9%). In short, this study implies that individual biomarkers could be limited in the differential diagnosis of lung cancer, and although the exploration of combined markers could be fruitful, tissue diagnosis should not be delayed. 

Yang et al. [64] conducted a study to diagnose and subtype lung cancer using serum protein profiles. They analyzed the sera of 123 lung cancer patients and 60 healthy subjects with surface enhanced laser desorption ionization (SELDI) ToF MS to build proteomic patterns. The data were statistically analyzed, and the differentiating performance of the protein mass-to-change ratio (M/Z) peaks was compared with ROC analysis. A total of 48 protein peaks were found to be differentially expressed in lung cancer patients compared to healthy individuals, out of which a subset of four (M/Z 1205, 4673, 1429, and 4279) showed variable expression in the three lung cancer subtypes. For SCC, there was an increase in the M/Z peaks at 1205, 4673, and 1429 and a decrease in the M/Z peak at 4279; for ADC, all four peaks were at the intermediate level, while SCLC showed the lowest levels of M/Z 1205 and 4673. M/Z 1205 exhibited the highest AUCs for distinguishing SCCs from SCLC and ADC (0.84 and 0.82, respectively). On the other hand, M/Z 4279 showed the best performance in differentiating SCLC and ADC (AUC = 0.87). An aggregate of M/Z 1205 and M/Z 4279 achieved the most superior AUC (0.91) for distinction between patients with SCCs and those with SCLCs. This study concluded that protein profiles could be used to facilitate the classification of lung cancer and could be especially helpful for patients with tumors in hard-to-reach anatomical sites or for those unable to undergo a tissue biopsy before treatment can start. 

Liang et al. [65] investigated the distinct glycosylation patterns of a serum glycoprotein called alpha 1 antitrypsin (A1AT) that could occur in different histological types of lung cancer. A1AT was extracted from serum samples of patients with ADC, SCC and SCLC and of benign controls (n = 48). Lectin microarrays were employed to identify three glycopatterns, which were later validated using a lectin-based ELISA. ROC analysis was utilized to assess the performance of each marker. Fucosylated A1AT could discern ADC from other subtypes (AUC = 0.844, specificity = 69%, sensitivity = 85.7%), while SCLC could be differentiated from NSCLC using poly-LacNAc levels (AUC = 0.707, specificity = 80.4%, sensitivity = 52.6%). Overall, this study revealed that specific glycopatterns of A1AT could be used as novel biomarkers for the classification of lung cancer.

Sugár et al. [66] aimed to classify lung cancer via proteomic techniques using nano UHPLC-MS (MS) analysis of a surface tryptic digestion of FFPE lung cancer tissue and adjacent tissue (n = 71) from 40 patients. There were varying numbers of differentially expressed proteins between the different histological types; 61 for ADC, 35 for SCC, and 99 for SCLC. Some examples included higher levels of eukaryotic translation initiation factor 1 and matrix-metalloproteinase proteins (MMP2 and MMP19) in ADC, upregulated fascin in SCC, and lower levels of IgG heavy constant gamma 2 and RAB10 in SCLC. The authors determined that there was a bigger difference in the proteomic characteristics between SCLC and NSCLC than among the subtypes of NSCLC. They concluded that proteomic profiles show promise as potential diagnostic markers. 

Table 3 tabulates a summary of proteomic studies and is headed with the following titles: study, sample size/type, marker, method, results, and conclusion. Highlights with the key findings are amended at the end as well.

### 4.3. Metabolomics

Rocha et al. [67] conducted a large-scale study (n = 100) in which they constructed a metabolomic profile of lung cancer using high-resolution magic-angle-spinning (HRMAS) NMR spectroscopy. The dataset was further analyzed using multivariate modeling and validation experiments. More significant increases were observed in certain metabolite levels in ADC and SCC subtypes compared to healthy controls (n PC, GPC, UDP/UTP, and peptides vs. lactate, glutamate, alanine, GSH, and creatine). Multivariate analysis yielded a classification rate of 81.3% for metabolomics when discriminating between ADC and SCC. The authors concluded that different subtypes of lung cancer exhibit distinct metabolic patterns, which means that NMR metabolomics could act as a valuable adjunct in lung cancer classification in the future. 

Similarly, Moreno et al. [68] also found apparent differences in metabolic footprints between the different histotypes of lung cancer. A total of 851 metabolites were isolated from 136 lung tissue samples using mass spectrometry techniques. Among these, nucleotides were consistently found to exhibit a higher trend in malignant tissue than in healthy tissue. 5,6-dihydrouracil was found to be particularly useful for discriminating ADC, and 20-O-methylguanosine and 5-methyluri-dine for SCC. 

Zang et al. [69] constructed metabolic profiles of 227 tissue samples from 79 lung cancer patients by employing ultra-performance liquid chromatography–high-resolution mass spectrometry (UPLC-HRMS). They identified 12, 4, and 7 metabolites with good accuracy, sensitivity, and specificity to distinguish ADC, SCC, and NSCLC from normal tissue. Valine, sphingosine, glutamic acid γ-methyl ester, and lysophosphatidylcholine (LPC) (16:0) made up the panel for ADC, and valine, sphingosine, LPC (18:1), and leucine derivatives were incorporated into the panel for SCC. A five-metabolite panel achieved 96.8% accuracy, 98.2% sensitivity, and 85.7% specificity when discriminating between ADC and SCC. 

You et al. [70] used a liquid chromatography–mass spectrometry-based non-targeted metabolomics analysis to build a metabolic profile of lung tissue from 131 patients. Overall, 241 metabolites were found to be differentially expressed in cancerous tissue. Moreover, logistic regression models identified a composite panel of creatine, myoinositol, and LPE 16:0 with good discriminatory power (AUC = 0.934) to distinguish between ADC and SCC. They concluded that these marker panels could serve as an excellent adjunct diagnostic tool for subtyping for lung cancer. 

In 2016, Klupczynska et al. [71] conducted the first ORBITRAP-based methodology (ultra-high-performance liquid chromatography–quadrupole-Orbitrap high-resolution mass spectrometry) to investigate biomarkers for lung cancer in the serum. After multiple validation steps, a total set of 12 unique metabolites was found to be significantly different in the cancer cohort, consisting of carnitine and two acylcarnitines (valerylcarnitine and propionylcarnitine), four amino acids (histidine, leucine, methionine, and tyrosine), two organic acids (pyroglutamic acid and malic acid), alpha-N-phenylacetyl-L-glutamine, thiomorpholine 3-carboxylate, and 1-amino-propan-2-ol/trimethylamine N-oxide/2-amino-1-propanol. An AUC of 0.836 was obtained by ROC analysis for this panel. However, this study could not elucidate a statistically significant panel difference in the distinction between ADC and SCC. 

Kowalczyk et al. [72] utilized liquid chromatography–mass spectrometry (LC-MS) for the metabolic profiling of tissue and plasma samples from 137 NSCLC patients. Their study revealed that tissue samples were a better medium for differentiating early stage ADC and SCC. Early stage SCC had higher levels of creatine, creatinine, xanthine, and dihydrothymine, whereas ADC had higher levels of fatty acids, carnitines, glycerophospholipids, lysoglycerophospholipids, amines, amino acids, and amides. For advanced stages of these diseases, however, serum samples were the better choice, with notable differences in fatty acids, carnitines, and fatty-acid amides. 

Yu et al. [73] used liquid chromatography coupled with tandem mass spectrometry (LC-MS/MS) on dried blood samples (DBSs) from 114 individuals to investigate the metabolic features of SCLC and NSCLC in male and female cohorts. ROC analysis resulted in an AUC of 95% for a five-metabolite panel (PI (18:0/18:0), Cer (d18:1/22:0 OH), 2-AG, IMP, and Cholic acid) for male patients with SCLC, while another panel (PE (18:1/20:4), 5-methyltetrahydrofolic acid, desmosterol, 4,5-dihydroorotic acid, and 9-HETE) yielded an AUC of 94% for female SCLC patients. Overall, this study proved that the metabolomic profiling of DBSs could be helpful in the diagnosis of SCLC. 

Qi et al. [74] attempted to employ metabolomic approaches to explore plasma metabolites as potential lung cancer biomarkers and prognostic markers based on their various subtypes. The plasma metabolomic profile of 173 individuals (98 lung cancer patients and 75 healthy controls) was obtained using high-resolution liquid chromatography–mass spectrometry (LC-MS). Certain metabolites (amino acids, fatty acids, and acylcarnitines) were found to be differentially expressed in lung cancer patients compared to controls. Further analysis using logistic regression models revealed panels of 5, 10, and 20 combinations of those markers to be distinctive between subtypes of lung cancer. A panel consisting of palmitic acid, heptadecanoic acid, ornithine, pentadecanoic acid, and acylcarnitine C8:1 proved to be the most effective subtype classifier. In short, the authors concluded that plasma metabolites could be a useful tool for the screening and classification of lung cancer, although it requires further validation. 

Cao et al. [75] employed LC coupled to electrospray ionization MS (LC-ESI-QTRAP-MS/MS) to identify the metabolomes of 128 plasma samples from patients with NSCLC, with the aim of distinguishing between ADC and SCC. In total, 1191 metabolites were found in samples, with 16 that were upregulated and 3 that were downregulated in ADC when compared to SCC. A panel consisting of 2-(methylthio) ethanol, cortisol, D-glyceric acid, and N-acetylhistamine were deemed as good differentiators between the two types, with an AUC of 0.946, 92.0% sensitivity, and 92.9% specificity. The authors concluded that their research could contribute to the growing evidence of metabolomic biomarkers as NSCLC discriminants.

Sieminska et al. [76] conducted a study on matched tissue and plasma samples from NSCLC patients, where they used LC-MS to construct the lipidome (derived from metabolomes) to discriminate between the two NSCLC subtypes. Eight lipid markers were found to be differentially expressed between ADC and SCC, namely upregulated plasma levels of LPA 20:4, LPA 18:1, and LPA 18:2; PC 16:0/18:2; OH, PC 18:0/20:4; OH, PC 16:0/20:4; and OOH and tumor levels of PC 16:0/18:2, PC 16:0/4:0, and CHO in ADC. ROC analysis revealed high AUCs for LPA 20:4 (0.851) and PC 16:0/20:4 (0.825). The diagnostic accuracy increased (AUC = 0.873) upon the combination of various markers. Thus, this research provides insight that oxPCs and LPAs could act as potential markers for ADC and SCC. 

Mazzone et al. [77] quantified various metabolites in 284 serum samples (94 from stage I–III NSCLC patients) and determined which metabolites differed significantly between the diseased and healthy subsets, along with the ratio of any differences. They used non-targeted UPLC-MS/MS and GC/MS techniques to design the study dataset. After correction for multiple factors, 149 metabolites were found to be differentially present in the lung cancer group compared to control. ADC patients showed different levels of 65 metabolites, out of which 5 were specific, while SCC patients had 50 metabolites with 3 specific ones that differed significantly from the control set. The study implies that with further exploration, the differences in the concentrations of various metabolites could be used as new biomarkers for NSCLC subtyping and to discover novel therapeutic targets. 

Yu et al. [78] studied the metabolic profiles of 143 blood samples with the objective of identifying metabolites that could differentiate between histotypes of lung cancer. They applied metabolomic techniques such as 1H NMR and UPLC-MS/MS to draw out a final set of 17 metabolites that could potentially be subtype specific. Further evaluation using stepwise discriminant analysis revealed five metabolites (serum xanthine, S-adenosylmethionine (SAM), carcinoembryonic antigen (CEA), neuron-specific enolase (NSE), and squamous cell carcinoma antigen (SCC)) as best suited for differentiation. A combined panel of these markers achieved a diagnostic accuracy of 92.3% and an AUC of 0.97 after ROC analysis. 

Shestakova et al. [79] used targeted metabolomic techniques to build the metabolic fingerprint of 200 plasma samples (100 from NSCLC patients and 100 from non-cancer patients) to determine the metabolites that differed significantly between the cohorts. Metabolites involved in the tryptophan metabolism, TCA cycle, urea cycle, and lipid metabolism pathways were found to be particularly distinctive. Further evaluation with partial correlation network analysis yielded metabolite ratios that could discriminate between the study groups. A model consisting of the candidate metabolites and their ratios exhibited an AUC value of 0.96. In short, this study implied that metabolomics integrated with bioinformatics could work as a valuable diagnostic tool for NSCLC. 

Table 4 summarizes studies exploring metabolomic markers. Key findings have been separately bulleted as highlights for the readers’ convenience.

## 5. Discussion

Lung cancer is one of the most prolific killers among cancers worldwide. It is of the utmost importance to accurately classify the histological subtypes of lung cancer to further the ongoing success of tailored treatment strategies while also balancing costs and patient comfort. This review aims at compiling and contrasting different omics methodologies that could potentially be integrated into the diagnostic flow for lung cancer subtyping. Similarities between studies evaluating individual techniques and markers have been drawn, and any discrepancies have also been addressed. This section ends with limitations and recommendations to minimize said limitations.

Several researchers have delved into the topic of omics data, providing updates on the recent development of various combinations of omics methods along with their clinical utility. For example, Bourbonne et al. [80] and Mei et al. [81] have explored the use of omics techniques in the prediction of treatment outcomes with immune checkpoint inhibitors. Abbasian et al. [82] have elaborated on proteomic and genomic biomarkers for the early detection and management of lung cancer, while Micheletti et al. [83] have provided a general overview of the biology behind multiple omics markers in regard to lung cancer diagnosis and treatment. To our best knowledge, our review is the first in its domain that delivers in-depth summaries of multiple studies that exclusively investigate the utilization of IHC, transcriptomics, proteomics, and metabolomics for lung cancer subtyping.

Among the diagnostic adjuncts discussed in this review, immunohistochemistry is the only one that has actually been adopted as a working member of the diagnostic schema for lung cancer subtyping. This technique comes with the advantages of successfully discriminating subtypes using small biopsy and cytological specimens, which eliminates the need for repeat biopsies and can also distinguish poorly differentiated tumors [19]. Various studies evaluating immunohistochemical markers appointed Napsin A and TTF-1 as good diagnostic markers for ADC and P40 and CK5/6 as effective SCC biomarkers. Both Zyl et al. [39] and Kim et al. [34] found that Napsin A was more specific and sensitive for ADC identification than TTF-1, while Tran et al. [42] achieved better results with P40 than its isotype p63 as an SCC marker. Certain authors deemed combined panels to be more effective than individual markers. Nishino et al. [36] obtained good results with a triple-marker panel consisting of TTF-1, Napsin A, and p40. Dual staining for p40/NapA and CK5/6/TTF1 was also effectively used by Guo et al. [38]. Kim et al. [34] elected a combined panel of Napsin A, TTF1, CK5/6, and p63 as the best option to differentiate between ADC and SCC. In contrast, Yassin et al. [35] and Roudi et al. [43] found an association between ALDH1 with NSCLC histotypes.

Although the studies reviewed in this section evaluated different miRNAs, most of the data indicate that transcriptomic markers could be potentially useful as adjunctive tools for histotype classification, except for those of Zhang et al. [49], whose study did not demonstrate a difference in miRNA values between ADC and SCC. Lu et al. [46] could observe differential levels of miR-17, miR-190b, and miR-375 between SCLC and NSCLC, although the further subdivision of NSCLC was not possible with this set. Various authors, including Yang et al. [48], Jin et al. [47], and Nadal et al. [54], constructed combined panels that achieved AUCs ranging from 0.89 to 0.99 for NSCLC diagnosis and subtyping. On the other hand, Saviana et al. [57] identified miR-375-3p as a specific diagnostic biomarker for SCLC. In general, all the studies concluded that transcriptomics could be promising in the field of adjunctive diagnosis for lung cancer.

Multiple studies have stipulated the helpful role that proteomics could play in lung cancer subtyping. CYFRA-21 and Pro-GRP were consistent among the better markers for NSCLC and SCLC diagnosis, respectively, as found by Visser et al. [59] and Korkmaz et al. [60], whereas Wen et al. [61] elected CEA and NSE as the most effective classifiers for SCLC. Furthermore, Trulson et al. [62] identified differential levels of SCC and CA15-3 between ADC and SCC. Korkmaz et al. [60] also drew a positive correlation between levels of CYFRA 21-1, HE4, and CgA and NSCLC stage. Several studies concurred that a combined panel of different proteomic markers performed better than individual ones as subtype classifiers. Among them, Zhuang et al. [58], Visser et al. [59], Korkmaz et al. [60], Sua et al. [63], and Trulson et al. [62] achieved high AUCs with models composed of but not limited to CYFRA-21, CEA, NSE, and Pro GRP.

Metabolomics has been used to study lung cancer subtypes, with several studies revealing distinct metabolic patterns. Many researchers have worked with a wide variety of metabolites, and some concordant marker sets have been agreed on, while there were contradictory experiments as well. Zang et al. [69], Kowalczyk et al. [72], Qi et al. [74], and Shestakova et al. [79] found differential levels of metabolites in the nucleotide, amino acid and fatty acid pathways between lung cancer patients and healthy controls, as well as between ADC and SCC, with varying degrees of AUC. Similarly, Moreno et al. [68] determined that nucleotides were the most effective distinguisher for subtype classification. In contrast, Jia et al.’s [84] study revealed that cancerous cells released more alcohols than non-cancerous cells and identified certain benzenes and phenols as the most efficient classifiers for NSCLC. Yu et al.’s [73] study was unique in the sense that they utilized dried blood samples from male and female SCLC patients and found varying levels of metabolites like IMP, cholic acid, desmosterol, etc., with high AUC values. Overall, most of the studies reviewed in this section concluded that metabolomics could work as an effective adjunct to lung cancer subtype classification, with the exception of Klupczynska et al. [71], who did not find any statistically significant difference in metabolites between subtypes.

Although each omics marker represents individual tiers in a multilevel schema of processes in a biological system, integrated approaches are necessary for a better understanding. So, in addition to the individual omics techniques mentioned above, there have been instances where researchers employed an integrated omics approach to lung cancer diagnostics. Ruiying et al. [85] formed a combined six-metabolite panel for NSCLC diagnosis that achieved an AUC of 0.99. The metabolomics results were further validated using transcriptomic techniques. This integrated approach displayed good comparability with diagnostic transcriptomes and metabolomes. Similarly, Stewart et al. [86] used a combined proteomic and metabolomic methodology to detect lung cancer, identifying 3891 peptides with differential expression in 18 SCLC and 18 NSCLC cell lines and over 100 metabolites as markers for differentiating subgroups.

It is worthy to note that patient variability and the ever-changing nature of proteins, as well as their insufficient availability in serum, are some challenges that need to be considered when employing large-scale proteomics [87]. In the case of metabolomics, the accurate identification of known and unknown compounds can prove to be tough due to the fine details of isotopes. This could be ameliorated by utilizing databases that house specific indicators of known compounds, i.e., accurate molecular mass and isotopic properties. Since the omics techniques handle such large and complicated datasets, smaller-scale studies are prone to have more experimental artifacts, as well as produce more false positives and be unreliable [88]. Complex mathematical, statistical, and bioinformatics tools are needed for data processing and integration [89]. The obstacles in biomarker discovery should also always be factored in while planning experimental designs. Improvements such as increasing sample size and using annotated samples and standard sample handling techniques will maximize the success of future studies.

This review is limited by its exclusion of radiomic and genomic techniques. Several recent studies have proven the high efficacy of radiogenomics in the field of lung cancer diagnosis. E et al. [90] obtained AUCs ranging from 0.665 to 0.822 between the various histological types with radiomic models. The ability of radiogenomic techniques to demonstrate EGFR expression and other transcriptional factors has also been shown by Zhou et al. [91].

The potential benefits of incorporating omics into the lung cancer diagnostic work flow warrant further studies to construct specific sets of standardized biomarkers so that clinicians can follow guidelines to direct management. Employing artificial intelligence to design sophisticated advanced bioinformatic tools will surely reduce some of the problems faced during data analysis [88]. Moreover, more studies need to be conducted investigating multifactorial effects on these biomarkers like age, gender, and smoking. The potential for an integrated approach should also be explored further to improve the accuracy and reliability of adjunct techniques.

## 6. Conclusions and Future Direction

In summary, this paper summarizes studies exploring immunohistochemical and “omics” techniques to highlight diagnostic biomarkers for lung cancer subtyping. Omics methodologies could prove to be a very valuable adjunctive tool in lung cancer diagnosis and subtyping. Incorporating these techniques into clinical practice has the potential to improve cost effectiveness and efficacy over existing conventional methods. However, utilizing omics markers on a large scale necessitates the construction of standardized biomarker sets that have been extensively studied and validated. Therefore, a more comprehensive effort needs to be undertaken to compile all the omics approaches to help design efficient multi-omics techniques and improve upon existing ones. Finally, the future of lung cancer diagnosis in the context of precision medicine lies with the integration of advanced, high-throughput multi-omics techniques and artificial intelligence.

## Figures and Tables

**Figure 1 cancers-16-02643-f001:**
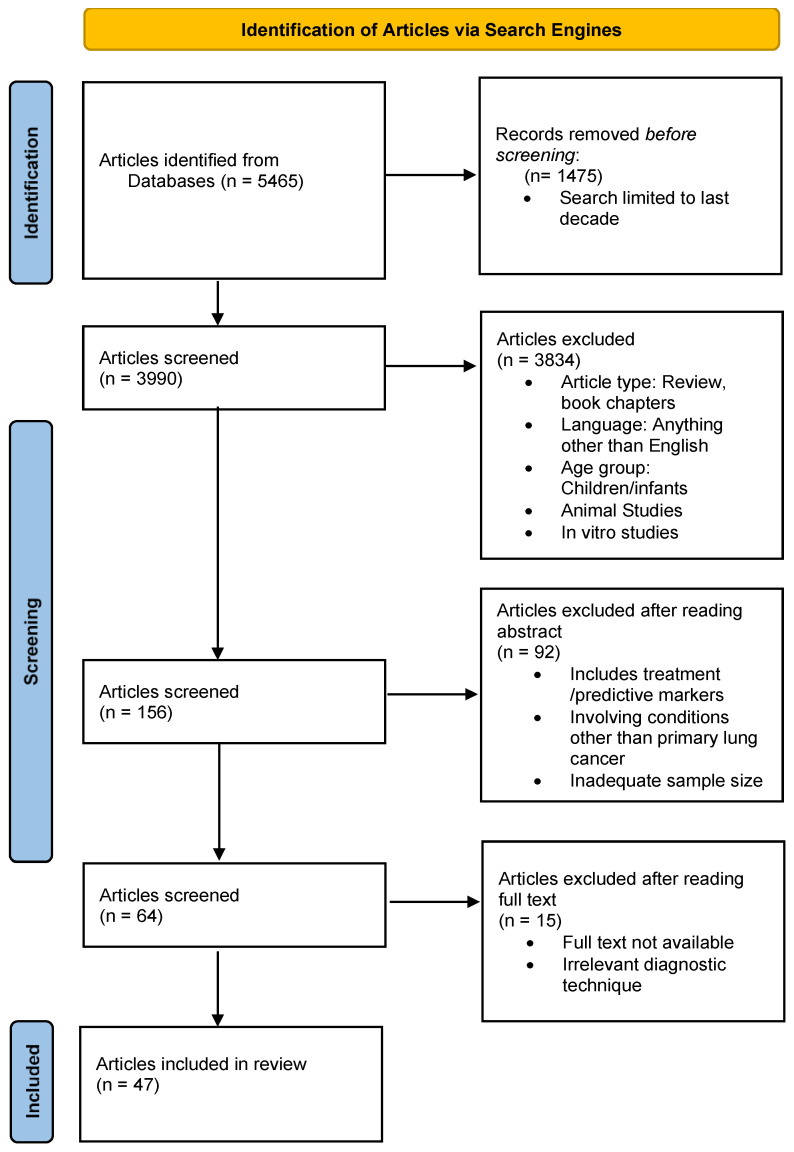
Schematic depiction of the article selection process.

**Table 1 cancers-16-02643-t001:** Summary of studies investigating immunohistochemical markers.

Study	Sample Size/Type	Markers	Methods	Results	Conclusion
Argon et al. [33]	120/ Lung tissue	TTF1, CK5/6, P63	TMA	Discriminator: TTF ○ Sensitivity: 100% ○ Specificity: 100% CK5/6 ○ Sensitivity: 97% ○ Specificity: 100% p 63 ○ Sensitivity: 100% ○ Specificity: 87%	TTF-1 is a reliable marker for subtyping lung cancer. CK5/6 and P63 can be used as adjuncts for final diagnosis.
Kim et al. [34]	129/ Lung, stomach, colon, vaginal tissue	Napsin A, TTF-1, p63, CK5/6 COX-2, HMWCK, CD141, p27, MOC-31, CEA, Rb protein	TMA	Discriminator: Napsin A was found to have a greater sensitivity and specificity than TTF1 (81% and 100% vs. 70% and 98%) for determining ADC. CK5/6 showed a sensitivity of 90% and specificity of 96% for determining SCC. P27, Rb, COX-2, HMWCK, MOC-31, and CEA might have limited value in the diagnosis of ADC.	Researchers elected a combined panel of Napsin A, TTF1, CK5/6, and p63 as the best option to differentiate between ADC and SCC.
Yassin et al. [35]	105/ Lung tissue	ALDH1A1	TMA	ADC: Overall, 83% of cases showed positive expression. ALDH1A1 expression was negatively correlated with the histological grade. SCC: Overall, 100% of the cases displayed positive expression. There was a negative correlation with histological grade. SCLC: Overall, 50% of cases had positive expression, higher immunoreactivity scores, and a more homogenous distribution in comparison to NSCLC sections.	There is a strong association between ALDH1 expression, histological subtype, and the grade of tumors in NSCLC.
Nishino et al. [36]	241/ Lung tissue	TTF1, NapA, CK5/6, and P40	TMA	SCC: p40 showed higher specificity to CK5/6 (94% vs. 59%, respectively). Sensitivity was 100% for both p40 and CK5/6). Overall, 100% of the SCCs were Napsin A−/p40− using the cocktail. ADC: TTF-1 displayed similar specificity to Napsin A (99% vs. 91%, respectively). Sensitivities (94% for both TTF-1 and napsin A) were similar. Overall, 87.3% were napsin A1+/p40− and none was napsin A−/p40−.	A cocktail of Nap A/p40 can be used to accurately subtype preoperative small biopsy and cytology specimens.
Wei et al. [37]	395/ Lung tissue	CDK5	TMA	CDK5 positivity in lung cancer samples highly exceeded that in normal lung tissue (51.5% vs. 20%). The positive rates were found to be higher in advanced cases, cases of lymphatic metastases, and cases with a poorer histological grade, AUC = 0.685.	CDK5 positivity is a good diagnostic marker for lung cancer.
Guo et al. [38]	58/ Lung tissue	p40/Nap A CK5/6/TTF1	TMA	100% of SCC cases showed p40 (nuclear) and CK5/6 (cytoplasmic) positivity, 93.8% of the ADC cases showed TTF1 (nuclear) positivity, and 87.5% cases showed Nap A (cytoplasmic) positivity. p40 and Napsin A dual-marker set showed 100% nuclear positivity and cytoplasmic negativity in SCC, while CK5/6 and TTF1 dual staining exhibited 100% cytoplasmic positivity and nuclear negativity in SCC. The single- and dual-marker panels revealed identical results.	Dual-marker stains could be an economical and simple novel approach to differentiating subtypes of NSCLC and could be an alternative to the more conventional single-marker IHC panels.
Zyl et al. [39]	271/ FNA and FFPE	TTF-1, Napsin A, CK5, and P63	TMA	ADC: Napsin A showed higher sensitivity than TTF1 (99.0% vs. 91.9%). Napsin A displayed a higher specificity than TTF-1 (90.8% vs. 62.8%). SCC: CK5 and P63 had similar sensitivity rates (95.4% vs. 97.9%) and NPVs. CK5 exhibited a greater specificity, PPV, and accuracy than P63.	Napsin A and TTF1 are acceptable markers for ADC, whereas CK5 and P63 are good markers for SCC.
Ao et al. [40]	200/ Lung tissue	Triple-marker panel (consisting of TTF-1, Napsin A, and p40	TMA	ADC: The triple marker showed higher sensitivity than TTF-1 and Napsin A alone (93.5% vs. 85.7% and 89.6%). Specificity was found to be lower for the triple marker than for Napsin A alone (77.5% vs. 90.0%). There was no statistically significant difference between the triple and single panels. SCC: Sensitivity Triple marker 88.3% P40 88.3% Ck5/6 89.6% P63 93.5% The triple-marker panel had higher specificity than p40, p63, and CK5/6 alone. There was no statistically significant difference between the panels.	The triple-marker panel could be very helpful for the accurate subclassification of NSCLCs and would also aid in tissue preservation for further necessary molecular testing.
Kawai et al. [41]	215/ Lung tissue	TTF1, Napsin A, p63, CK5/6, desmocollin-3	TMA	Discriminators: Napsin A ○ Sensitivity: 80% ○ Specificity: 97% p63 ○ Sensitivity: 84% ○ Specificity: 98%	Napsin A and P63 could be very helpful in the histological categorization of NSCLC.
Tran et al. [42]	557/ Lung tissue	CK5, p63, p40, Napsin A, TTF-1	TMA	IHC markers of SCC identification and NSCLC exclusion: CK5 and p40 p40 showed superior diagnostic performance to P63. Markers of exclusion of SCC: NAP A and TTF1 TTF-1 clone SPT24 was slightly more sensitive and less specific than clone 8G7G3/1.	CK5 and p40 are good diagnostic markers for SCC and superior to p63.
Roudi et al. [43]	133/ Lung tissue	ALDH1 CD133	TMA	ALDH1 levels were higher in NSCLC cases than SCLC cases. Of the NSCLC cases, SSC had the highest ALDH1 expression. Greater levels of CD133 were found in NSCLC cases compared to SCLC cases. Significantly different levels were not determined between the different NSCLC subtypes. ADC and SCC cases exhibited high levels of both markers, while all other tumor types displayed high levels of one marker and lower levels of the other marker.	ALDH1 and CD133 could be used as novel biomarkers for lung cancer subtyping.
Roudi et al. [44]	195/ Lung tissue	CD44	TMA	NSCLC cases exhibited a greater expression of CD44 than SCLC cases. Among the NSCLC subtypes, SCC showed higher levels of CD44 compared to ADC. Higher levels of the same marker corresponded with higher-grade SCC tumors with poorer prognosis. Lower levels indicated well-differentiated ADC tumors.	CD44 could potentially serve as a novel diagnostic biomarker and might also be used to formulate targeted therapies.

Highlights: (1) Tran et al. [43] achieved better results with P40 than its isotype p63 as an SCC marker. (2) Nishino et al. [36] obtained good results with a triple-marker panel consisting of TTF-1, Napsin A, and p40. (3) Dual stains p40/NapA and CK5/6/TTF1 were used by Guo et al. [38]. (4) Kim et al. [34] elected a combined panel of Napsin A, TTF1, CK5/6, and p63 as the best option to differentiate between ADC and SCC. (5) Yassin et al. [35] and Roudi et al. [43] found an association between ALDH1 and NSCLC histotypes. (6) Zyl et al. [39] found that Napsin A was more specific and sensitive for ADC identification than TTF-1.

**Table 2 cancers-16-02643-t002:** Summary of studies investigating transcriptomic techniques.

Study	Sample Size/Type	Marker	Method	Results	Conclusion
Geng et al. [45]	50/ Blood	miR-20a, miR-223, miR-21, miR-221, miR-145	RT-PCR	Best candidates: miR-20a (AUC = 0.89) miR-223 (AUC = 0.94) miR-155 (AUC = 0.92)	These five miRNAs could individually work as noninvasive biomarkers for early stage NSCLC.
Lu et al. [46]	1132/ Plasma	Six miRNAs	qRT-PCR	miR-17, miR-190b, and miR-375 could discriminate between SCLC and NSCLC. Training set AUC = 0.878 Validation set AUC = 0.869 The subdivision of NSCLC was not possible.	These panels could potentially serve as effective markers for the early diagnosis and subtyping of lung cancer.
Jin et al. [47]	78/ Macrodissected frozen lung tissue	miR-365	qRT-PCR	miR-365 was upregulated in SCLC and ADC. miR-365 was downregulated in SCC.	miR-365 is higher in SCLC and ADC and lower in SCC.
Yang et al. [48]	175/ Blood	Eight circulating miRNAs	qRT-PCR	A panel of four miRNAs (miR-146b, miR-205, miR-29c, miR-30b, and miR-337) was best for the early diagnosis of NSCLC. For ADC: AUC = 0.98 Sensitivity: 99.1% For SCC: AUC = 0.93 Sensitivity: 90.3%	This panel can be utilized for the early diagnosis of NSCLC and is a better predictor for ADC than SCC.
Zhang et al. [49]	147/ Serum	16 miRNAs	qRT-PCR	NSCLC patients had significantly higher miR-3149 and miR-4769.3p (AUC 0.830 and 0.735, respectively). The values were similar in both ADC and SCC.	miR-3149 and miR-4769.3p performed well as diagnostic markers for NSCLC.
Powrozek et al. [50]	90/ plasma	miR-944 miR-3662	RT-PCR	Both miRNAs were higher in cancerous samples compared to the non-cancer cohort. miR-944 had a greater accuracy for operable SCC (AUC = 0.982), and miR-3662 for ADC (AUC = 0.926).	miR-944 and miR-3662 could serve as potent biomarkers for lung cancer diagnosis in the future.
Singh et al. [51]	80/ Blood	Six miRNAs	qRT-PCR	mir-2116 and mir-449c were upregulated only in AC. mir-2115, mir-548q, and mir-2117 were higher only in SCC. mir-2115 and mir-449c were highest in SCC and AC, respectively.	These markers (specifically mir-2115 and mir-449c) could be effective in the differential diagnosis of NSCLC within the scope of noninvasive approaches.
Kumar et al. [52]	115/ Serum	10 miRNAs	RT-PCR	miR-15a-5p, miR-320a, miR-25-3p, let-7e-5p, miR-192-5p, let-7d-5p miR-148a-3p, and miR-92a-3p were lower in NSCLC patients. SCC patients had lower levels of miR-375 and miR-10b-5p compared to controls. There was no significant difference in these markers between ADC and controls.	miRNA-based assays could assist in the diagnosis and prognosis of NSCLC.
Kumar et al. [53]	161/ Tissue samples	miR-375-3p miR-197-3p	TaqMan advanced miRNA assays	miR-375-3p and miR-197-3p were upregulated in tumor resectates compared to healthy controls. miR-375-3p was upregulated in tissue samples. miR-375-3p was the best diagnostic marker AUC = 0.7491 miR-375-3p and miR-197-3p were good differentiators of SCC and ADC.	Lung carcinogenesis might be linked with the excessive expression of miR-375-3p and miR-197-3p. The expression of miR-375-3p may assist in predicting therapeutic response.
Nadal et al. [54]	94/ Serum	Four miRNAs	qRT-PCR	A combined panel could diagnose NSCLC. AUC = 0.993	This miRNA signature could act as a clinically helpful tool for the determination of lung cancer in patients.
Jin et al. [55]	88/ Serum	MiRNA	RNA sequencing	ADC patients had significantly upregulated levels of 11 and downregulated levels of 13 miRNAs (miR-181a-5p, miR-30a-3p, miR-30e-3p, and miR-361-5p were specific for ADC). SCC patients had upregulated levels of six and lower levels of eight miRNAs (miR-10b-5p, miR-15b-5p, and miR-320b for SCC). The three-miRNA combined panel was a good diagnostic marker for NSCLC, ADC, and SCC, with AUCs of 0.899, 0.936, and 0.911, respectively.	The three-miRNA combined panel showed promise as a good diagnostic marker for NSCLC, ADC, and SCC
Fan et al. [56]	152/ Plasma	12 miRNAs	qRT-PCR fluorescence quantum dot liquid bead array	Six miRNAs (miR-16-5p, miR-20a-5p, miR-92-3p, miR-15b-5p, miR-17b-5p, miR-19-3p) were able to distinguish between the two groups. miR-15b-5p, miR-16-5p, and miR-20a-5p were the best discriminants for NSCLC.	These six markers could be explored as an avenue for noninvasive NSCLC diagnosis.
Saviana et al. [57]	144/ Plasma	Seven circulating cell-free miRNAs	RNA sequencing qRT-PCR	miR-375-3p is the most competent discriminant for SCLC. AUC = 0.793 Sensitivity: 37.5% Specificity: 100%	Cell-free miRNA circulation shows promise as low cost, noninvasive, and reliable biomarker for SCLC diagnosis.

Highlights: (1) Lu et al. [46] could identify differential levels of miR-17, miR-190b, and miR-375 between SCLC and NSCLC (AUC = 0.869), although the further subdivision of NSCLC was not possible. (2) Yang et al. [48], Jin et al. [55], and Nadal et al. [54] constructed combined panels that achieved AUCs ranging from 0.89 to 0.99 for NSCLC diagnosis and subtyping. (3) Zhang et al. [49] demonstrated no significant difference in MiRNA values between ADC and SCC. (4) Saviana et al. [57] identified miR-375-3p as a specific diagnostic biomarker for SCLC (AUC = 0.793).

**Table 3 cancers-16-02643-t003:** Summary of studies investigating proteomic markers.

Study	Sample Size/Type	Marker	Method	Results	Conclusion
Yu et al. [58]	180/ Serum	CEA NSE CPR LDH Cyfra 21-1	Mass spectrometry	GEP 3 (CEA + NSE + Cyfra 21-1) Accuracy: 94.8%	A panel consisting of CEA, NSE, and CYFRA 21-1 was found to be superior for classifying lung cancer.
Visser et al. [59]	1096/ Blood	Eight protein tumor markers (CA125, CA15.3, CEA, CYFRA 21-1, HE4, NSE, proGRP, SCCA) ctDNA mutations in EGFR, KRAS, and BRAF	Electrochemiluminescent assays Droplet digital PCR	CYFRA 21-1 had the best overall performance in identifying NSCLC. AUC = 0.78 sensitivity: 25% Pro GRP was best for identifying SCLC. AUC = 0.86 Sensitivity: 40% PPV: 100%. A model combining CYFRA 21-1, CEA, proGRP, and NSE performed better than CYFRA 21-1 alone for NSCLC diagnosis. AUC = 0.86 The panel of CA125, CA15.3, CYFRA 21-1, NSE, proGRP, and DNA was superior to individual TMs when diagnosing SCLC.	These models have clinical value and may help in LC diagnostics.
Korkmaz et al. [60]	129/ Venous blood	Six tumor markers: ProGRP SCCAg CYFRA 21.1 HE4 CgA NSE	Mass spectrometry	ProGRP levels were found to be higher in SCLC. CYFRA 21.1 and SCCAg levels were higher in NSCLC. CYFRA 21.1 (*p* < 0.001, r = 0.394), HE4 (*p* = 0.014, r = 0.279), and CgA (*p* = 0.023, r = 0.259) levels had a positive correlation with the stage of NSCLC. Pro GRP was the best discriminator between subtypes. AUC = 0.875	A panel of three tumor markers, CYFRA 21.1, HE4, and ProGRP, may play a role in differentiating LC from benign lung disease as well as subtyping SCLC.
Wen et al. [61]	250/ Blood	CEA, CYFRA 21, NSE	Mass spectrometry	The CEA level was the highest in ADC. AUC = 0.812 Sensitivity: 63.9% CYFRA 21 (AUC: 0.847; sensitivity: 84.6%) and CEA (AUC: 0.804; sensitivity: 70.0%) were the best suited for SCC. NSE (AUC: 0.819; sensitivity: 69.0%) and CEA (AUC: 0.808; sensitivity: 60.7%) were the most related to SCLC.	Tumor markers could diagnose different lung cancer types.
Trulson et al. [62]	490/ Serum	CYFRA 21-1 CEA, NSE ProGRP SCC, CA 125 CA 15-3 CA 19-9 CA 72-4	Mass spectrometry	SCLC vs. NSCLC Pro GRP and NSE AUCs were 0.86 and 0.83, respectively. A two-marker panel (NSE and ProGRP) and a three-marker combination (NSE, ProGRP, and CEA) performed better; AUC = 0.97. ADC vs. SCC Low SCC and high CA15-3 AUC = 0.75	A combination of serum tumor markers could be useful in assessing different subtypes of lung cancer.
Sua et al. [63]	93/ Serum	Pro-GRP, CEA, CYFRA 21-1, SCC-Ag, and NSE	Mass spectrometry	Median levels of Pro-GRP, CEA, CYFRA 21-1, SCC-Ag, and NSE were higher in malignant lung disease than benign lung disease Discriminators: NSCLC Panel of Pro-GRP, CEA, CYFRA 21-1, SCC-Ag, and NSE AUC = 81.1% Sensitivity = 74.5% Specificity = 78.8% SCLC CYFRA 21-1 + Pro-GRP + NSE AUC = 97.3% Sensitivity = 88.8% Specificity = 98.9%	Individual biomarkers could be limited in the differential diagnosis of lung cancer, and although the exploration of combined markers could be fruitful, tissue diagnosis should not be delayed.
Yang et al. [64]	183/ Serum	Protein mass-to-change ratio (M/Z) peak	SELDI-TOF-MS	Overall, 48 protein peaks were differentially expressed in lung cancer patients compared to healthy individuals. For SCC, there was an increase in the M/Z peaks at 1205, 4673, and 1429 and a decrease in the peak M/Z 4279. For ADC, M/Z 1205, 4673, 1429, and 4279 were at the intermediate levels and showed the lowest levels of M/Z 1205 and 4673. Best discriminators: SCCs vs. SCLC and ADC M/Z 1205 (AUC 0.84 and 0.82, respectively) SCLC vs. ADC M/Z 4279 AUC = 0.87 SCCs vs. SCLC Combination of M/Z 1205 and M/Z 4279 AUC = 0.91	Protein profiles could be used to facilitate the classification of lung cancer and could be especially helpful for patients with tumors in hard-to-reach anatomical sites or for those unable to undergo a tissue biopsy before treatment can start.
Liang et al. [65]	48/ Serum	Glyco patterns of A1AT	Lectin microarray Lectin-based ELISA	Discriminator: ADC vs. subtypes Fucosylated A1AT AUC = 0.844 Specificity = 69%, Sensitivity = 85.7% SCLC vs. NSCLC Poly-LacNAc AUC = 0.70 Specificity = 80.4% Sensitivity = 52.6%	Specific glycopatterns of A1AT could be used as novel biomarkers for the classification of lung cancer.
Sugar et al. [66]	71/ Lung tissue	Number of differentially expressed proteins	Nano UHPLC-MS (MS)	Number of differential proteins: ADC: 61 (eukaryotic translation initiation factor 1 and matrix-metalloproteinase proteins (MMP2 and MMP19)) SCC: 35 (Fascin) SCLC: 99 (IgG heavy constant gamma 2 and RAB10)	Proteomic profiles show promise as potential diagnostic markers.

Highlights: (1) Visser et al. [59] and Korkmaz et al. [60] found that CYFRA-21 (AUC = 0.78) and Pro-GRP were consistent among the better markers for NSCLC and SCLC diagnosis, respectively. (2) Wen et al. [61] elected CEA (AUC = 0.812) and NSE (AUC = 0.819) as the most effective classifiers for SCLC. (3) Trulson et al. [62] identified differential levels of SCC and CA15-3 between ADC and SCC. (4) Korkmaz et al. [60] drew a positive correlation between levels of CYFRA 21-1, HE4, and CgA and NSCLC stage. (5) Zhuang et al. [58], Visser et al. [59], Korkmaz et al. [60], Sua et al. [63], and Trulson et al. [62] achieved high AUCs (0.811–0.97) with models composed of but not limited to CYFRA-21, CEA, NSE, and Pro GRP.

**Table 4 cancers-16-02643-t004:** Summary of studies investigating metabolomic methods.

Study	Sample Size/Type	Marker	Method	Results	Conclusion
Rocha et al. [67]	100/ Plasma	n PC, GPC, UDP/UTP, and peptides vs. lactate, glutamate, alanine, GSH, and creatine	HRMAS NMR spectroscopy	Higher levels of n PC, GPC, UDP/UTP, and peptides were observed in ADC. Higher levels of lactate, glutamate, alanine, GSH, and creatine were observed in SCC. Accuracy: 81.3%	Different subtypes of LC exhibit distinct molecular patterns.
Moreno et al. [68]	136/ Lung tissue	Nucleotides	Mass spectrometry	5,6-dihydrouracil was discriminatory for ADC and 20-O-methylguanosine and 5-methyluri-dine for SCC.	Lung cancer histotypes can be determined by metabolomic profiling
Zang et al. [69]	227/ Lung tissue	Valine, sphingosine, glutamic acid γ-methyl ester, lyso phosphatidylcholine (LPC) (16:0), LPC (18:1), leucine derivatives	UPLC-HRMS	Discriminators ADC: Valine, sphingosine, glutamic acid γ-methyl ester, and lysophosphatidylcholine (LPC) (16:0) SCC: Valine, sphingosine, LPC (18:1), and leucine derivatives For a five-metabolite panel Accuracy: 96.8% Sensitivity: 98.2% Specificity: 85.7%	This panel shows a promising prospect for NSCLC tissue detection and subtyping.
You et al. [70]	131/ Lung tissue	241 metabolites	LC-MS	Panel of creatine, myoinositol, and LPE 16:0 AUC = 0.934	This panel could serve as an excellent adjunct diagnostic tool for subtyping for lung cancer.
Klupczynska et al. [71]	75/ Serum	Amino acids (histidine, leucine, methionine, and tyrosine), organic acids (pyroglutamic acid, malic acid), carnitine, acylcarnitines (valerylcarnitine and propionylcarnitine), alpha-N-phenylacetyl-L-glutamine, thiomorpholine 3-carboxylate, 1-amino-propan-2-ol/trimethylamine N-oxide/2-amino-1-propanol	ORBITRAP-based methodology Ultra-high-performance liquid chromatography–quadrupole- Orbitrap high- resolution mass spectrometry	AUC = 0.836 for discrimination between lung cancer and normal tissue No statistically significant difference was observed between subtypes.	Orbitrap-based global metabolic profiling could be a useful diagnostic and therapeutic strategy in NSCLC.
Kowalczyk et al. [72]	137/ Lung tissue and plasma	Amino acids, fatty acids, carnitines, lyso glycerophospholipids, sphingomyelins, plasmalogens glycerophospholipids, metabolites related to N-acyl ethanolamine (NAE) biosynthesis, glycerophospho (N-acyl) ethanolamines (GP-NAE)	LC-MS	Tissue samples were a better medium for differentiating early stage ADC and SCC. Serum samples were better for advanced disease- Early stage SCC had higher levels of creatine, creatinine, xanthine, and dihydrothymine. ADC had greater levels of fatty acids, carnitines, glycerophospholipids, lyso glycerophospholipids, amines, amino acids, and amides. Fatty acids, carnitines, and fatty-acid amides were discriminatory for advanced cases.	Metabolites can differentiate between NSCLC histotypes.
Yu et al. [73]	114/Dried blood	Metabolites (PI (18:0/18:0), Cer (d18:1/22:0 OH), 2-AG, IMP, Cholic acid, PE (18:1/20:4), 5-methyltetrahydrofolic acid, desmosterol, 4,5-dihydroorotic acid, 9-HETE)	LC-MS/MS	SCLC Male: (PI (18:0/18:0), Cer (d18:1/22:0 OH), 2-AG, IMP, and cholic acid AUC = 0.95 Female: PE (18:1/20:4), 5-methyltetrahydrofolic acid, desmosterol, 4,5-dihydroorotic acid, and 9-HETE AUC = 0.94	Metabolomic profiling of DBSs could be helpful in the diagnosis of SCLC.
Qi et al. [74]	173/Plasma	Metabolites (Amino acids, fatty acids, acylcarnitines)	High-resolution LC-MS	Palmitic acid, heptadecanoic acid, ornithine, pentadecanoic acid, and acylcarnitine C8:1 were the most effective subtype classifiers.	Plasma metabolites could be a useful tool for the screening and classification of lung cancer.
Cao et al. [75]	128/Plasma	1191 metabolites, including lipids	LC-ESI -QTRAP- MS/MS	Number of differential metabolites between ADC and SCC: 16 upregulated 3 downregulated Best discriminators: 2-(methylthio) ethanol, cortisol, D-glyceric acid, and N-acetylhistamine AUC = 0.946 Sensitivity = 92.0% Specificity = 92.9%	Their research could contribute to the growing evidence of metabolomic biomarkers as NSCLC discriminants.
Sieminska et al. [76]	122/Lung tissue 101/Plasma	Lipid markers	LC-MS	Discriminants: ADC Upregulated plasma levels of LPA 20:4, LPA 18:1, and LPA 18:2; PC 16:0/18:2; OH, PC 18:0/20:4; OH, PC 16:0/20:4; OOH Upregulated tumor levels of PC 16:0/18:2, PC 16:0/4:0, CHO Best performers: LPA 20:4 (AUC = 0.851) PC 16:0/20:4 (AUC = 0.825) Combined panel (AUC = 0.873)	This research provides insight that oxPCs and LPAs could act as potential markers for ADC and SCC.
Mazzone et al. [77]	284/Serum	Serum metabolites	Non-targeted UPLC-MS/MS and GC/MS	Overall, 149 metabolites were differentially present in the lung cancer group compared to the controls. ADC patients showed different levels of 65 metabolites (5 specific). SCC patients showed different levels of 50 metabolites with 3 specific ones.	The differences in the concentrations of various metabolites could be used as new biomarkers for NSCLC subtyping and to discover novel therapeutic targets.
Yu et al. [78]	143/Blood	Various metabolites	1H NMR UPLC- MS/MS	SAM, CEA, NSE, and SCC were best suited for differentiation. Accuracy = 92.3% AUC = 0.97	The model can be used to discriminate between lung cancer subtypes.
Shestakova et al. [79]	200/Plasma	Amino acids Tryptophan metabolism intermediates Acylcarnitine	Targeted metabolomic techniques	Metabolites involved in the tryptophan metabolism, TCA cycle, urea cycle, and lipid metabolism pathways were distinctive. Model of combined markers and ratios: AUC = 0.96	Metabolomics integrated with bioinformatics could work as a valuable diagnostic tool for NSCLC.

Highlights: (1) Zang et al. **[69]**, Kowalczyk et al. [72], Qi et al. [74], and Shestakova et al. [79] found differential levels of metabolites in the nucleotide, amino acid, and fatty acid pathways between lung cancer patients and healthy controls, as well as between ADC and SCC, with varying degrees of AUC. (2) Moreno et al. [68] reported that nucleotides were the most effective distinguishers for subtype classification. (3) Yu et al. [73] found varying levels of metabolites like IMP, cholic acid, desmosterol, etc., with high AUC (0.94–0.96) values in male and female SCLC patients. (4) Klupczynska et al. [71] did not find any statistically significant difference in metabolites between subtypes.

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
