# Peer review of "Lung Cancer Subtyping: A Short Review"

_cancers, 2024, doi:10.3390/cancers16152643_

Round 1

Reviewer 1 Report

Comments and Suggestions for Authors

In this study, the authors tried to discuss the lung cancer subtypes. Some potential points need to be addressed;

 1) Some original studies that used the TMA method for lung cancer markers such as ALDH1, CD133 and CD44, and so on are missing including Zahra Madjd, et al 2014 and 2015. The authors should re-do the literature search and include all the related articles!

2) A few comprehensive review articles published on the topic such as Mohammad Hadi Abbasian, et al, 2020. The author should use this article and similar ones and clarify the novelty of their work.

Comments on the Quality of English Language

minor English edit 

Reviewer 2 Report

Comments and Suggestions for Authors

The outcomes and curation of biomarkers for Lung Cancer is contributing and welcoming, however it requires severe restructuring and revision to align the outcomes to match with the title, methods and conclusions. Here are my suggestions:

1. Language and structure of summary and abstract require improvement to aid towards understanding of the purpose of reviewing lung cancer subtyping as it might have already in the domain. Additionally, the summary should clearly mention how the review is contributing to demonstrate that it is different and advanced than the existing one. A typical review summary pattern is suggested to be followed.2. Heading 1: Authors have abbreviated Lung Cancer as LC which appears redundant as it hasn't been used anywhere in the manuscript except as LC/MS that contradicts the whole manuscript. By the abbreviation LC/MS authors don't mean Lung Cancer/Mass Spectroscopy. Also, LC as an abbreviation for Lung Cancer should be discouraged against NSCLC, SCLC etc which are already well known and understood.3. Heading 2: Authors are abruptly mixing meta analysis while attempting to review entirely different concepts of subtyping from biomarker from omics data. Here 6,786 articles as results with the given set of words  are skeptical as author's have not mentioned the exclusion and inclusion criteria apart from the set of words. Additionally, authors should mention the tool or software involved in the analysis or research.Overall, authors need thorough revision of the manuscript to restructure it as the detailed outcomes doesn't match with the title, summary, and conclusion of the manuscript.

Comments on the Quality of English Language

Minor revision for English language

Reviewer 3 Report

Comments and Suggestions for Authors

The authors describe quite well what is the global scenario of lung cancer. They focus a lot on the "omics" applications and the state of the art in the sector. Nevertheless I believe that an in-depth focus on liquid biopsies, as new generation themes, with the perspectives that derive from them can and must be integrated to define such a complex and varied picture. I would invite to add some other publications referring to the use of liquid biopsies in lung cancer.

Furthermore: in fig.1 there is an arrow that leads to nowhere. I would suggest removing it.

Round 2

Reviewer 2 Report

Comments and Suggestions for Authors

The paper is significantly improved with suggested revisions. It can be accepted in its present form.